# High mortality among kidney transplant recipients diagnosed with coronavirus disease 2019: Results from the Brazilian multicenter cohort study

**Lúcio R. Requião-Moura**[1,2,3‡]*, **Tainá Veras de Sandes-Freitas**[4,5,6‡], **Laila Almeida Viana**[2], **Marina Pontello Cristelli**[2], **Luis Gustavo Modelli de Andrade**[7], **Valter Duro Garcia**[8], **Claudia Maria Costa de Oliveira**[5], **Ronaldo de Matos Esmeraldo**[6], **Mario Abbud Filho**[9], **Alvaro Pacheco-Silva**[3], **Katia Cronemberger Sousa**[10], **Alessandra Rosa Vicari**[11], **Kellen Micheline Alves Henrique Costa**[12], **Denise Rodrigues Simão**[13], **Marcos Vinicius de Sousa**[14], **Juliana Bastos Campos**[15], **Ricardo Augusto Monteiro de Barros Almeida**[7], **Luciane Mônica Deboni**[16], **Miguel Moysés Neto**[17], **Juliana Aparecida Zanocco**[18], **Helio Tedesco-Silva**[1,2‡], **José Medina-Pestana**[1,2‡], *on behalf of COVID-19-KT Brazil*[¶]

**1** Department of Medicine, Nephrology Division, Federal University of São Paulo, São Paulo, SP, Brazil, **2** Department of Transplantation, Hospital do Rim, Fundação Oswaldo Ramos, São Paulo, SP, Brazil, **3** Renal Transplant Unit, Hospital Israelita Albert Einstein, São Paulo, SP, Brazil, **4** Department of Clinical Medicine, Federal University of Ceará, Fortaleza, CE, Brazil, **5** Hospital Universitário Walter Cantídio, Fortaleza, CE, Brazil, **6** Hospital Geral de Fortaleza, Fortaleza, CE, Brazil, **7** Department of Internal Medicine, Universidade Estadual Paulista-UNESP, Botucatu, SP, Brazil, **8** Santa Casa de Misericórdia de Porto Alegre, Porto Alegre, RS, Brazil, **9** Hospital de Base, Medical School FAMERP, São José do Rio Preto, SP, Brazil, **10** Federal University of Maranhão, São Luiz, MA, Brazil, **11** Hospital de Clínicas de Porto Alegre, Federal Univertisy of Rio Grande do Sul, Porto Alegre, RS, Brazil, **12** Division of Nephrology and Kidney Transplantation, Onofre Lopes University Hospital, Natal, RN, Brazil, **13** Hospital Santa Isabel, Blumenau, SC, Brazil, **14** Division of Nephrology, Renal Transplant Unit, Renal Transplant Research Laboratory, School of Medical Sciences, University of Campinas–UNICAP, Campinas, SP, Brazil, **15** Santa Casa de Misericórdia de Juiz de Fora, Juiz de Fora, MG, Brazil, **16** Hospital Municipal São José (HMSJ), Joinville, SC, Brazil, **17** Division of Nephrology, School of Medicine of Ribeirão Preto, University of Sao Paulo, Ribeirão Preto, SP, Brazil, **18** Hospital Santa Marcelina, São Paulo, SP, Brazil

‡ Lúcio R. Requião-Moura and Tainá Veras de Sandes-Freitas are Co first authors on this work. Also Helio Tedesco-Silva and José Medina-Pestana are Co senior authors on this work.
¶ Membership of the COVID-19-KT Brazil is listed in the Acknowledgments.
* lucio.moura@einstein.br

## Abstract

### Background

Kidney transplant (KT) recipients are considered a high-risk group for unfavorable outcomes in the course of coronavirus disease 2019 (COVID-19).

### Aim

To describe the clinical aspects and outcomes of COVID-19 among KT recipients.

### Methods

This multicenter cohort study enrolled 1,680 KT recipients diagnosed with COVID-19 between March and November 2020, from 35 Brazilian centers. The main outcome was the

**Data Availability Statement:** All relevant data are within the manuscript and its Supporting Information files.

**Funding:** The present study was partially supported by Novartis Pharma Brazil. This support was not associated with any relevant declarations relating to employment, consultancy, patents, products in development and marketed products. Novartis Brazil, represented by the Evidence Generation Medical Affairs Pharma Team, established a research collaboration with TVSF to support the execution of the study. This collaboration involved financial support to operational costs with study execution, statistical data analysis, language review and submission fees. In addition, Novartis collaborated with protocol and final study report review, support with the regulatory process review and data cleaning. There was no additional funding received from Novartis Brazil for this study. There was no additional external funding received for this study.

**Competing interests:** The present study was partially supported by Novartis Pharma Brazil. This does not alter our adherence to PLOS ONE policies on sharing data and materials.

**Abbreviations:** AKI, acute kidney injury; AZA, azathioprine; BMI, body mass index; CNI, calcineurin inhibitor; COVID-19, coronavirus disease 2019; DD, deceased donor; eGFR, estimated glomerular filtration rate; ICU, intensive care unit; IRB, Institutional Review Board; IS, immunosuppressive; KAL, kidney after liver; KT, kidney transplant; LD, living donor; mTORi, mammalian target of rapamycin inhibitor; MV, mechanical ventilation; MW, Midwest; NE, northeast; PCR, polymerase chain reaction; PAK, pancreas after kidney; RRT, renal replacement therapy; S, south; SARS-CoV-2, severe acute respiratory syndrome coronavirus 2; SE, southeast; SHK, simultaneous heart-kidney; SLK, simultaneous liver-kidney; SPK, simultaneous pancreas-kidney.

90-day cumulative incidence of death, for the entire cohort and according to acute kidney injury (AKI) and renal replacement therapy (RRT) requirement. Fatality rates were analyzed according to hospitalization, intensive care unit (ICU) admission, and mechanical ventilation (MV) requirement. Multivariable analysis was performed by logistic regression for the probability of hospitalization and death.

## Results

The median age of the recipients was 51.3 years, 60.4% were men and 11.4% were Afro-Brazilian. Comorbidities were reported in 1,489 (88.6%), and the interval between transplantation and infection was 5.9 years. The most frequent symptoms were cough (54%), myalgia (40%), dyspnea (37%), and diarrhea (31%), whereas the clinical signs were fever (61%) and hypoxemia (13%). Hospitalization was required in 65.1%, and immunosuppressive drugs adjustments were made in 74.4% of in-hospital patients. ICU admission was required in 34.6% and MV in 24.9%. In the multivariable modeling, the variables related with the probability of hospitalization were age, hypertension, previous cardiovascular disease, recent use of high dose of steroid, and fever, dyspnea, diarrhea, and nausea or vomiting as COVID-19 symptoms. On the other hand, the variables that reduced the probability of hospitalization were time of COVID-19 symptoms, and nasal congestion, headache, arthralgia and anosmia as COVID-19 symptoms. The overall 90-day cumulative incidence of death was 21.0%. The fatality rates were 31.6%, 58.2%, and 75.5% in those who were hospitalized, admitted to the ICU, and required MV, respectively. At the time of infection, 23.2% had AKI and 23.4% required RRT in the follow-up. The cumulative incidence of death was significantly higher among recipients with AKI (36.0% vs. 19.1%, P < 0.0001) and in those who required RRT (70.8% vs. 10.1%, P < 0.0001). The variables related with the probability of death within 90 days after COVID-19 were age, time after transplantation, presence of hypertension, previous cardiovascular disease, use of tacrolimus and mycophenolate, recent use of high dose of steroids, and dyspnea as COVID-19 symptom. On the other hand, the variables that reduced the risk of death were time of symptoms, and headache and anosmia as COVID-19 symptoms.

## Conclusion

The patients diagnosed with COVID-19 were long-term KT recipients and most of them had some comorbidities. One in every five patients died, and the rate of death was significantly higher in those with AKI, mainly when RRT was required.

## Introduction

In March 2020, about 3 months after the first confirmed case in China, coronavirus disease 2019 (COVID-19), the infectious disease caused by the novel coronavirus severe acute respiratory syndrome coronavirus 2 (SARS-CoV-2), was declared a pandemic by the World Health Organization. In Brazil, the first case was registered on February 26, 2020. To date, Brazil is one of the most affected countries, with > 220,000 deaths in 9.2 million infected people by January 2021 [1]. Two relevant aspects of this pandemic could be highlighted, not only in Brazil but also worldwide: geographic differences in the spread of the virus and a wide variety of

clinical presentations in infected patients [2–5]. In Brazil, a country with a continental dimension and diverse economic and climatic variations, the initial spread of the pandemic followed an evident migratory character, affecting different geographic areas at different times, with varied intensity, and with each geographic region showing different waves in the number of cases across the epidemiologic weeks [6]. Even in the federative unit of São Paulo, which is the most populous state in the country, it was possible to detect this pattern of migratory movements [7].

Clinically, it has been estimated that the vast majority of patients present with an asymptomatic picture or a discreet flu syndrome; however, others can evolve to severe presentations associated with severe acute respiratory syndrome, requiring mechanical ventilation (MV) and having a high risk of death [4]. To date, age has been well known to be one of the strongest predictors of death, although comorbidities such as hypertension, diabetes, and obesity have contributed to the worst prognosis [5, 8]. Accordingly, kidney transplant (KT) recipients have been considered a group of patients with a high vulnerability to infection and worst outcomes, not only because of the chronic and unavoidable use of immunosuppressive drugs but also owing to the presence of comorbidities related to a severe evaluation [9–12].

Considering these scenarios, it is imperative to measure the impact of the COVID-19 pandemic in this specific population, despite the evolution of knowledge related to the diagnosis, improvement and amplification of testing and refinement of patient care throughout 2020 [13]. Therefore, this study aimed to describe the characteristics of COVID-19 in KT recipients followed up in different Brazilian transplantation centers, located in different geographic areas, over the first month of the pandemic in the country, as well as to measure the main outcomes after the COVID-19 diagnosis in these patients.

## Methods

### Study design and population

This was a retrospective multicenter cohort study that enrolled KT recipients followed up in centers located in four of the five geographic regions in Brazil. The country is geographically divided into five regions: north, northeast, midwest, southeast, and south. All 81 active KT centers in Brazil were invited, of which 78 agreed to participate and 37 effectively completed the regulatory process. The centers in the north region did not answer the invitation or did not complete the regulatory process; therefore, none of the enrolled patients came from centers located in this region. The 35 participating centers are responsible for 57% of all transplantation activities in the country. The study was approved by the National Ethics Research Committee (CAEE 30631820.0.1001.8098) and by the local ethics committee of all participating centers (S1 Table), and it was registered in the Clinical.Trails.gov (NCT04494776). Informed consent or its exemption followed specific national legislations, the local Institutional Review Board (IRB) recommendations, and the guidelines of the Declaration of Helsinki. In all cases where informed consent was obtained, it was written. In some specific cases, the informed consent was waived by the local IRB.

The eligible participants were KT recipients or KT combined with other solid organ recipients who underwent transplantation performed at any time, of any age, and with a COVID-19 diagnosis between March and November 11, 2020. The diagnosis was considered only in patients who presented at least one COVID-19-attributable symptom associated with a positive result in any one of the following tests: polymerase chain reaction (PCR), serology, or viral antigen detection. A codebook with the list of the attributable symptoms is presented in the S2 Table. The diagnosis based on serology was retrospectively performed in patients who had had attributable symptoms. The attributable symptoms were defined by the local investigator.

Screening diagnoses in asymptomatic patients were excluded. The final follow-up date was December 11, 2020, or the date of death, and the analysis was performed between December 2020 and January 2021.

## Source of data

All participating centers completed an *ad hoc* designed data collection form containing demographic, epidemiologic, clinical, laboratory, clinical management option, and outcome data. Data were anonymized and de-identified, and stored in the REDCap platform. One physician (TVSV) checked the quality of all data.

## Definitions and outcomes

The provenance of the cases was considered from the location of the center where the patients were being followed up and was grouped according to the geographic region of the country: northeast, midwest, southeast, and south. The source of infection was classified as either community or probably nosocomial (defined as an infection diagnosis during hospitalization for a clinical condition not related to COVID-19), according to the clinical criteria of each center. Immunosuppressive management, use of medications for infection, patients' allocation to the ward or intensive care unit (ICU), and indication for MV were performed at each center according to their own local practices. The place of clinical management was categorized as domiciliary or in-hospital. Patients whose clinical management was started at home but were hospitalized in the course of the infection were considered to have had in-hospital clinical management. The clinical management of patients who were allocated to home care was defined by the local investigators according to their own practices.

The baseline creatinine value was assessed from the mean value of the three last available serum creatinine measurements before the date of infection, and the estimated glomerular filtration rate (eGFR) was calculated using the Chronic Kidney Disease Epidemiology Collaboration equation [14]. Acute kidney injury (AKI) was considered present if the serum creatinine level at the COVID-19 diagnosis was higher by $\geq 50\%$ than the baseline value [15]. Renal replacement therapy (RRT) was indicated according to local practices.

The main outcome was the 90-day cumulative incidence of death after the COVID-19 diagnosis. The intermediate outcomes were need for hospitalization because of COVID-19 infection, ICU admission, MV requirement, AKI at COVID-19 diagnosis, and RRT requirement during hospitalization.

## Statistical analysis

Categorical variables are summarized as frequency distributions and percentiles. Continuous variables were tested for normality using the Kolmogorov-Smirnov and Shapiro-Wilk tests. As no one variable presented a normal distribution, all of them are described as median and interquartile range (first to third quartiles). Missing values were censored, and the number of non-missing values was included in the description of the variables. The fatality rates (proportion of deaths from COVID-19 compared to the number of diagnosed patients) were calculated for the whole population and for groups according to intermediate outcomes: hospitalization, ICU admission, MV requirement, AKI at COVID-19 diagnosis, and RRT requirement.

The outcomes were grouped according to the time after transplantation, dichotomized in $\leq 1$ year or longer. Additionally, all variables were compared between hospitalized and non-hospitalized patients, as well as survivors and non-survivors patients. The continuous variables were compared by Mann-Whitney U test and the categorical were compared by $X^2$ or Fisher test. The variables that reached a P-value $\leq 0.10$ in the univariable comparison were

selected for the multivariable modeling for the probability of hospitalization and death. The multivariable analysis was performed by the binary logistic regression.

The cumulative incidences of death were calculated for the whole population and according to AKI and RRT requirement using Kaplan-Meier analysis, and compared using the log rank test. Statistical analyses were performed using Statistical Package for the Social Sciences (version 26; IBM, Armonk, NY, USA), and statistical significance was defined as $P < 0.05$, with the 95% confidence interval.

## Results

### Demographic data and case distribution over time

The demographic data at the COVID-19 diagnosis are detailed in Table 1. The median age of the recipients was 51.3 (41.7–60.0) years, and 75.1% of them were aged between 18 and 60 years. Most of the KT recipients were men (60.4%), and only 11.4% were Afro-Brazilian. A high prevalence of comorbidities was observed, with hypertension being the most frequent (75.7%), followed by diabetes mellitus (34.0%) and obesity (23.8%), although the median body mass index was 26.5 (23.6–29.7) kg/m$^2$. Some previous cardiovascular diseases were reported in 12.3% of the patients, and no comorbidity was registered in 11.4%. Reflecting the demography of the Brazilian transplant program, 66.7% of the patients received a graft from deceased donors, 31.2% from living donors, and only 2.2% received KT simultaneously or sequentially with another solid organ. The time interval between transplantation and COVID-19 infection was 5.9 (2.3–10.7) years. The most frequent immunosuppressive regimen that patients were receiving at the time of infection was the combination of one calcineurin inhibitor (tacrolimus or cyclosporin) with mycophenolate (59.4%), followed by one calcineurin inhibitor with azathioprine (15.5%). Use of high dose of steroids and lymphocyte depleting antibody to treat acute rejection until 3 months before the COVID-19 diagnosis were recorded for 4.3% (n = 73) and 2.8% (n = 47), respectively. Only 8.7% of the patients were receiving a steroid-free regimen. The eGFR was 48.4 (32.4–65.4) mL/min/1.73 m$^2$.

Over time, a relevant variation was observed in terms of the number of cases according to the geographic area, as shown in Fig 1. Across all months, the southeast region had the highest number of cases, especially between June and July (208 and 205 cases, respectively). The same trend was observed in the northeast region, which showed an earlier peak of 137 diagnoses in May, followed by a large decrease from August and with only one case registered in November. Meanwhile, the south region recorded an increase in cases from June, reaching a peak of 87 diagnoses in July, followed by a plateau between September and November.

### Source of infection and clinical management

The main source of infection was the community, which was reported in 92% of the patients. Nosocomial transmission was most frequently reported in the first 2 months (March and April), with an incidence reaching 14% of all cases in each month, followed by 12% in July and 6–7% in the other months. Among the patients who had nosocomial transmission, in 16 (11.7%) the infection occurred during the hospitalization when the transplantation was performed. The time between the start of symptoms and the COVID-19 diagnosis was 5 (3–9) days, and their frequencies are depicted in Fig 2. The most frequent symptoms were cough (54%), myalgia (40%), dyspnea (37%), and diarrhea (31%). Notably, a quarter of the patients had anosmia (24%) and only 9% presented with ageusia. Among the clinical signs, fever or chills were observed in 61% and hypoxemia in 13%. The diagnoses were made according to the results of PCR test in 84.6%, serology test in 15.3%, and viral antigen detection in 0.8%. Among patients who had the diagnosis based on serology, the time between the COVID-19

**Table 1. Demographic characteristics at COVID-19 diagnosis.**

|  | Non-missing cases | Total N = 1,680 |
|---|---|---|
| Age (years) | 1,680 | 51.3 (41.7–60.0) |
| Age group | 1,680 | |
| *< 18 years* | | 14 (0.8) |
| *18–60 years* | | 1261 (75.1) |
| *> 60 years* | | 405 (24.1) |
| Male sex | 1,680 | 1,015 (60.4) |
| Afro-Brazilian ethnicity | 1,680 | 191 (11.4) |
| BMI (kg/m$^2$) | 1,588 | 26.5 (23.6–29.7) |
| BMI $\geq$ 30 kg/m$^2$ | 1,588 | 378 (23.8) |
| Donor source | 1,680 | |
| *KT-LD* | | 524 (31.2) |
| *KT-DD* | | 1,120 (66.7) |
| *SPK* | | 23 (1.4) |
| *PAK* | | 1 (0.1) |
| *SLK* | | 7 (0.4) |
| *KAL* | | 4 (0.2) |
| *SHK* | | 1 (0.1) |
| Time after KT (years) | 1,676 | 5.9 (2.3–10.7) |
| KT $\leq$ 30 days | 1,676 | 41 (2.4) |
| Comorbidities | 1,680 | |
| *None* | | 191 (11.4) |
| *Hypertension* | | 1,272 (75.7) |
| *Diabetes* | | 571 (34.0) |
| *Cardiovascular* | | 206 (12.3) |
| *Pulmonary* | | 54 (3.2) |
| *Neurologic* | | 20 (1.2) |
| *Hepatic* | | 63 (3.8) |
| *Neoplasia* | | 84 (5.0) |
| *Autoimmune* | | 49 (2.9) |
| IS regimen | 1,680 | |
| *CNI-AZA* | | 261 (15.5) |
| *CNI-MPA* | | 998 (59.4) |
| *CNI-mTORi* | | 157 (9.3) |
| *CNI-free* | | 165 (9.8) |
| *Other* | | 99 (5.9) |
| eGFR (mL/min/1.73 m$^2$) | 1,497 | 48.4 (32.4–65.4) |

AZA, azathioprine; BMI, body mass index; CNI, calcineurin inhibitor; DD, deceased donor; eGFR, estimated glomerular filtration rate; IS, immunosuppressive; KAL, kidney after liver; KT, kidney transplant; LD, living donor; mTORi, mammalian target of rapamycin inhibitor; PAK, pancreas after kidney; SHK, simultaneous heart-kidney; SLK, simultaneous liver-kidney; SPK, simultaneous pancreas-kidney.

attributable symptoms and serology was 26.0 (9.0; 55.0) days. In 12 patients, more than one diagnostic method was reported.

About two-thirds of the patients (65.1%) were hospitalized for clinical management, 7.0 (4.0; 10.0) days after the beginning of symptoms. Among them, the main therapeutic management was the prescription of antimicrobials (Table 2): 56.5% received azithromycin and 70.7%

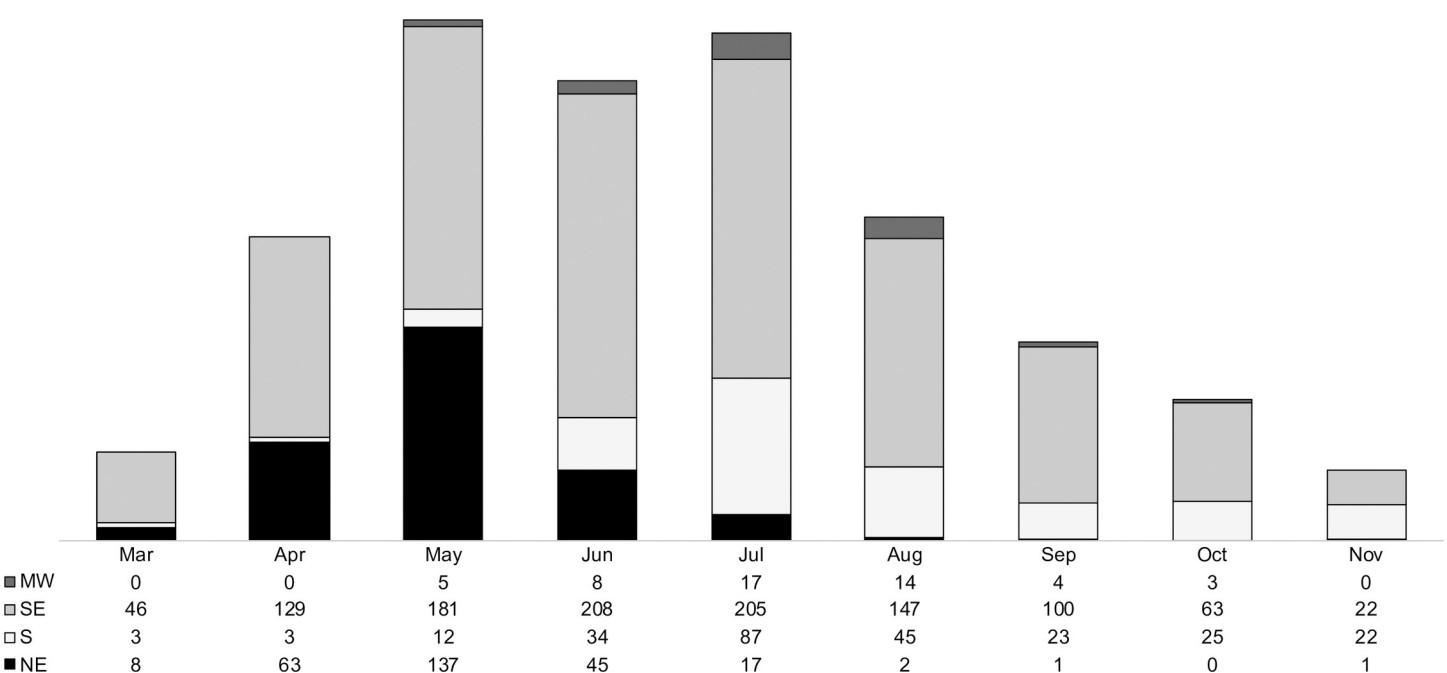

**Fig 1. Number of COVID-19 cases over the observation months according to geographic region.** MW, midwest; SE, southeast; S, south; NE, northeast.

received antibiotics other than azithromycin. In 16.6% of the patients, the antiviral oseltamivir was prescribed. Increase in the steroid dose was reported in 43.6% of the patients. Hydroxy-chloroquine or chloroquine was prescribed in 16.0%; however, its use was predominant in the first 2 months of the pandemic (March and April), abruptly decreased from May (7%), ranged from 2.5% to 1.4% in the following months, and was not reported in November. Some antico-agulants were indicated in prophylactic doses in 29.5% of in-hospital patients, whereas only 8.8% received therapeutic doses.

Another important point was the management of the immunosuppressive schedule. In 74.4% of in-hospital patients, some kind of intervention in immunosuppressive drugs was indicated, with a reduction in the dose of the antiproliferative drug (mycophenolate, azathio-prine, or mammalian target of rapamycin inhibitor) being the choice in 37.2%, whereas both antiproliferative drugs and calcineurin inhibitors were withdrawn in 36.4%. As expected, a very different profile was observed among the 34.9% patients who were allocated to home care (Table 2). Among them, 32.9% received azithromycin and 14.2% ivermectin. A high-dose ste-roid was prescribed in 12.5% of the patients. In the vast majority (84.0%), there was no change in the immunosuppressive schedule. In 14.8%, the dose of the antiproliferative agent was reduced.

## Outcomes

The creatinine value at baseline and at the time of COVID-19 diagnosis was reported in 1,497 and 1,071 patients, respectively. Therefore, in 1,052 recipients (62.6% of the entire cohort), the creatinine values were available at the two time points, and the frequency of AKI at COVID-19 diagnosis was 23.2% (244 patients). ICU admission was required in 34.6% (n = 577, missing values = 11) and MV in 24.9% (n = 417, missing values = 10). The intermediate outcomes are summarized in Table 3. The cumulative incidence of death at 90 days after the diagnosis was 21.0% (Fig 3). The fatality rate was grouped according to intermediate outcomes (Table 3).

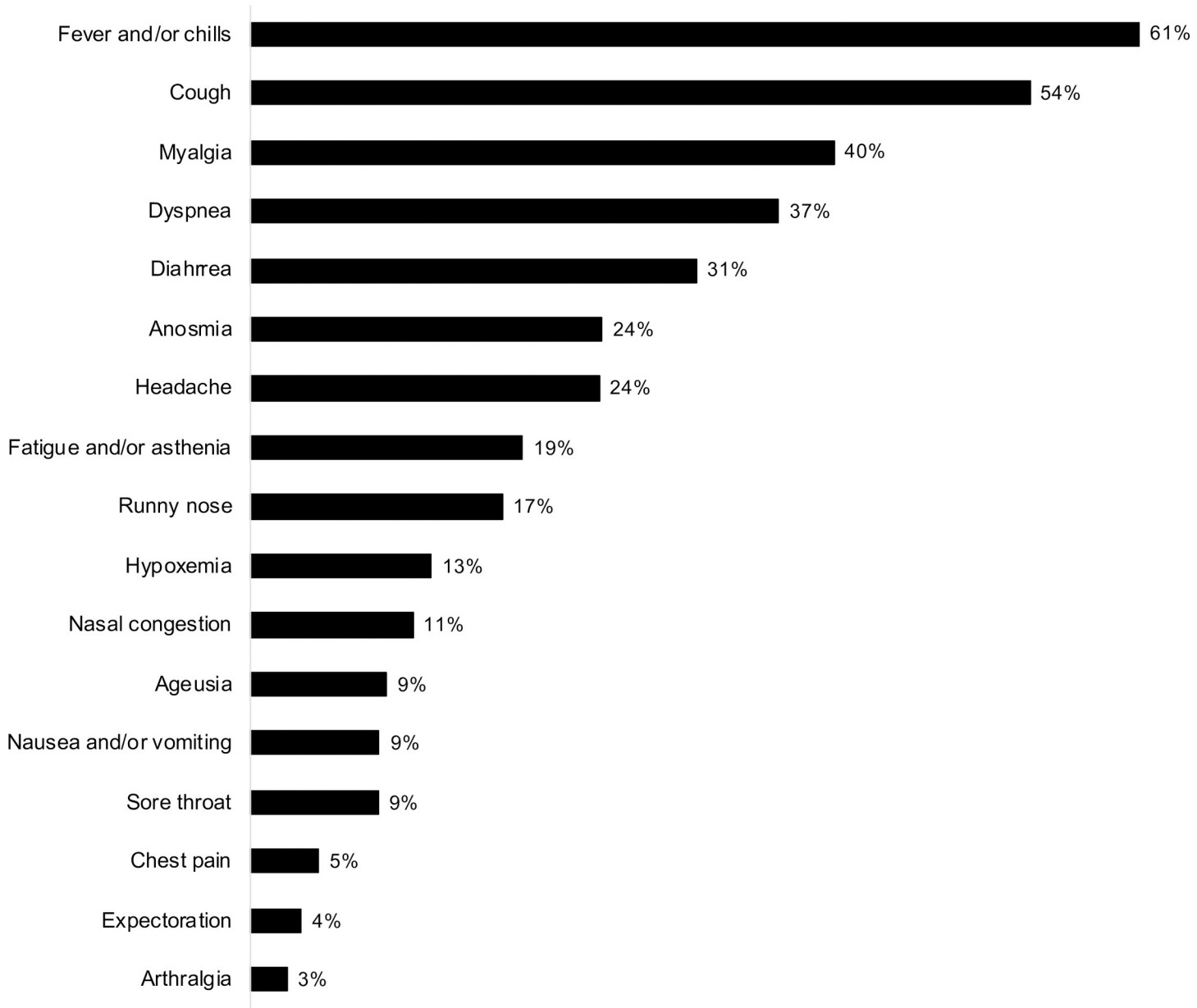

**Fig 2. Symptoms and signs at COVID-19 diagnosis.**

The fatality rate was 31.6% among in-hospital patients, 58.2% (n = 336) in those who were admitted to the ICU, and 75.5% (n = 314) in those who required MV. The outcomes were compared according to the time of transplantation: ≤ 1 year or longer (S3 Table). The hospitalization rate was higher among patients who had the time after transplantation shorter than 1 year (71.3% vs. 64.0%, P = 0.031), as well as the requirement for RRT (30.6% vs. 22.1, P = 0.005).

We observed that the cumulative incidence of death at 90 days after the COVID-19 diagnosis was significantly higher in patients who had AKI (36.0 vs. 19.1%, P < 0.0001; Fig 4A) and in those who required RRT during the hospitalization time (70.8% vs. 10.1%, P < 0.0001; Fig 4B).

**Table 2. Clinical management of in-hospital patients and outpatients.**

| Clinical management, n (%) | In-hospital n = 1,094 (65.1%) | Outpatient n = 589 (34.9%) |
|---|---|---|
| Supportive medicines[a] | | |
| Azithromycin | 618 (56.5) | 193 (32.9) |
| Antibiotics (other than azithromycin) | 774 (70.7) | 92 (15.7) |
| High-dose steroids | 477 (43.6) | 73 (12.5) |
| Anticoagulants | | |
| Prophylactic dose | 324 (29.5) | 11 (1.9) |
| Therapeutic dose | 96 (8.8) | 2 (0.3) |
| (Hydroxy)chloroquine | 175 (16.0) | 16 (2.7) |
| Oseltamivir | 182 (16.6) | 133 (5.6) |
| Ivermectin | 102 (9.3) | 83 (14.2) |
| Nitazoxanide | 5 (0.5) | 8 (1.4) |
| Management of immunosuppressive schedule | | |
| Antiproliferative drug reduction or withdrawal | 385 (37.2) | 87 (14.8) |
| CNI reduction or withdrawal | 48 (4.4) | 5 (0.9) |
| Complete withdrawal | 398 (36.4) | 1 (0.2) |
| No change | 280 (25.6) | 492 (84.0) |

CNI, calcineurin inhibitor.

Footnote:

[a] The sum of variables is > 100% because the patients could have received more than one clinical management option.

We compared the variables among patients who were hospitalized with those who were not. The results are presented in the S4 Table. The variables that achieved better results were included in a multivariable analysis, which is detailed in the Table 4. The variables related with the probability of hospitalization were age, hypertension, previous cardiovascular disease, recent use of high dose of steroid, and fever, dyspnea, diarrhea, and nausea or vomiting as COVID-19 symptoms. On the other hand, the variables that reduced the probability of hospitalization were time of COVID-19 symptoms, and nasal congestion, headache, arthralgia and anosmia as COVID-19 symptoms.

Finally, We compared the variables among patients who survived with those who did not. The results are presented in the S5 Table. The variables that achieved better results were included in a multivariable analysis, which is detailed in the Table 5. The variables related with the probability of death within 90 days after COVID-19 were age, time after transplantation, presence of hypertension, previous cardiovascular disease, use of tacrolimus and mycophenolate as the maintenance immunosuppression, recent use of high dose of steroids, and dyspnea as COVID-19 symptom. On the other hand, the variables that reduced the risk of death were time of symptoms, and headache and anosmia as COVID-19 symptoms.

**Table 3. Fatality rates according to intermediate outcomes.**

| Intermediate outcomes | Non-missing cases | Frequency n (%) | Fatality rate n (%) |
|---|---|---|---|
| Hospitalization | 1,680 | 1,094 (65.1) | 346 (31.6) |
| ICU requirement | 1,669 | 577 (34.6) | 336 (58.2) |
| MV requirement | 1,670 | 417 (24.9) | 314 (75.7) |
| AKI | 1,052 | 244 (23.2) | 86 (35.2) |
| RRT | 1,670 | 319 (23.4) | 273 (69.8) |

AKI, acute kidney injury; ICU, intensive care unit; MV, mechanical ventilation; RRT, renal replacement therapy.

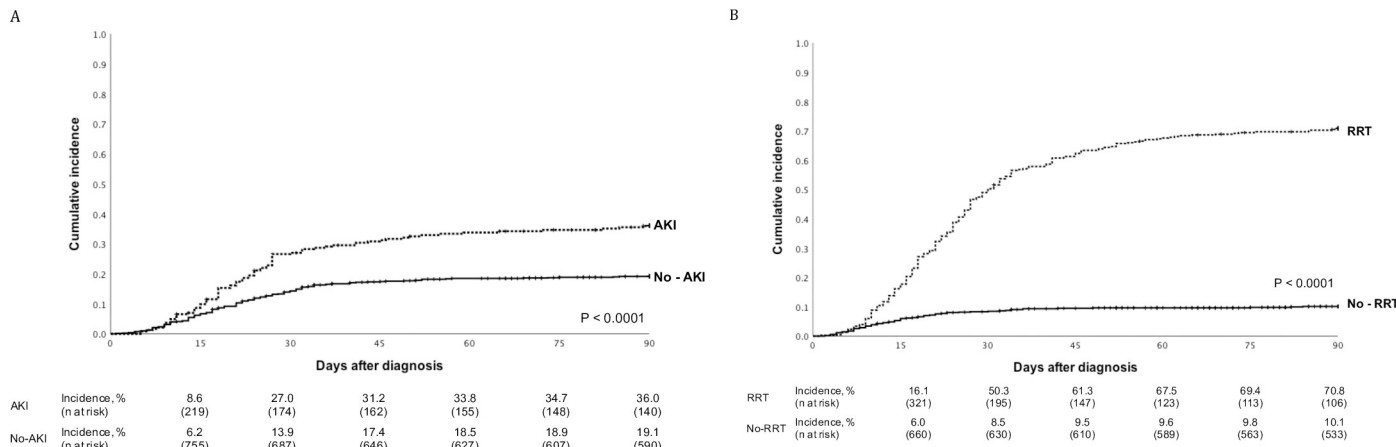

| Overall | Incidence, % (n at risk) | 6.1 (1,680) | 15.1 (1,572) | 18.5 (1,323) | 20.0 (1,261) | 20.6 (1,200) | 21.0 (1,141) |
|---|---|---|---|---|---|---|---|

**Fig 3. Ninety-day cumulative incidence of death.**

**Fig 4. Ninety-day cumulative incidence of deaths according to AKI and RRT requirement.** A: Cumulative incidence of deaths according to the presence of AKI at COVID-19 diagnosis. AKI was considered to be present in patients whose serum creatinine at COVID-19 diagnosis was increased by 50% compared to level at baseline (mean of three last available serum creatinine measurements). B: Cumulative incidence of deaths according to RRT requirement during hospitalization. AKI, acute kidney injury; COVID-19, coronavirus disease 2019; RRT, renal replacement therapy.

**Table 4. Multivariable analysis for the probability of hospitalization.**

| Variables | OR | 95% CI | P-value |
|---|---|---|---|
| Age (for each year) | 1.033 | 1.023–1.044 | <0.001 |
| Hypertension (yes vs. no) | 1.418 | 1.076–1.869 | 0.013 |
| Cardiovascular disease (yes vs. no) | 1.648 | 1.077–2.523 | 0.021 |
| Recent high dose of steroids (yes vs. no) | 1.866 | 1.229–2.832 | 0.003 |
| Time of symptoms (for each day) | 0.983 | 0.971–0.996 | 0.008 |
| Fever (yes vs. no) | 1.649 | 1.290–2.108 | <0.001 |
| Dyspnea (yes vs. no) | 3.673 | 2.803–4.812 | <0.001 |
| Nasal congestion (yes vs. no) | 0.588 | 0.410–0.845 | 0.004 |
| Headache (yes vs. no) | 0.525 | 0.402–0.685 | <0.001 |
| Arthralgia (yes vs. no) | 0.250 | 0.117–0.533 | <0.001 |
| Diarrhea (yes vs. no) | 1.783 | 1.367–2.325 | <0.001 |
| Anosmia (yes vs. no) | 0.519 | 0.387–0.696 | <0.001 |
| Nausea and vomiting (yes vs. no) | 2.284 | 1.427–3.655 | 0.001 |

AKI, acute kidney injury; eGFR, estimated glomerular filtration rate.

Footnote: Variables included in the modeling: age, type of donor (living vs. deceased); hypertension, diabetes, previous cardiovascular disease, and neoplasia as comorbidities; recent use of high dose of steroids and recent use of Thymoglobulin; time of the start of COVID-19 symptoms; fever, chills, cough, dyspnea, runny nose, headache, nasal congestion, sore throat, myalgia, arthralgia, diarrhea, nausea and vomiting, anosmia, and ageusia as COVID-19 symptoms.

Values with missing values < 10%: recent use of high dose of steroids and recent use of Thymoglobulin. For both the mode value was imputed, i.e., no use.

Variables excluded from the modeling:

 a) number of missing were higher than 10%: eGFR (10.9%) and AKI (37.4%)

 b) collinearity: hypoxemia (colinear with dyspnea).

## Discussion

In this multicenter study that enrolled KT recipients being followed up in 35 Brazilian transplant centers, which together account more than half of the national transplantation activities, we presented a general panorama of the impact of COVID-19 in a cohort of 1,680 recipients diagnosed with the disease, which is the largest reported cohort to date. The spread of SARS-CoV-2 infection followed an evident migratory character, affecting different geographic regions with varied intensities, as summarized in Fig 1. Brazil is a continental country with geographic regions showing very different demographic densities, in addition to deep economic and social inequalities. Moreover, the distribution of the number of cases throughout the observation months in 2020 in this study was similar to the pattern observed in a large Brazilian study that included the first 250,000 Brazilians diagnosed with COVID-19 who required in-hospital clinical management [6].

In this study, the main source of transmission was the community; however, 8% of the patients had a probable nosocomial transmission. In the first cohort followed in Wuhan, China, there was a clear concern about establishing the primary source of contact, with special efforts to describe the natural history of transmission [4, 5, 16]. Therefore, at the beginning of the pandemic, approximately 2% of the infected patients had a reported contact with wildlife [4] and it was possible to determine the community as the source of infection in 40% of in-hospital patients [5]. As expected, in the largest population studies that were subsequently published, the source of contamination has been mostly the community, except among healthcare workers in whom the source was unavoidable occupational exposure [17]. However,

**Table 5. Multivariable analysis for the probability of death.**

| Variables | OR | 95% CI | P-value |
|---|---|---|---|
| Age (for each year) | 1.054 | 1.040–1.067 | <0.001 |
| Time after transplantation (for each year) | 1.025 | 1.002–1.047 | 0.030 |
| Hypertension (yes vs. no) | 1.566 | 1.070–2.293 | 0.021 |
| Cardiovascular disease (yes vs. no) | 1.517 | 1.047–2.198 | 0.028 |
| CNI-MPA (vs. others) | 1.197 | 1.022–1.401 | 0.026 |
| Recent high dose of steroids (yes vs. no) | 1.534 | 1.063–2.214 | 0.022 |
| Time of symptoms (for each day) | 0.954 | 0.928–0.981 | 0.001 |
| Dyspnea (yes vs. no) | 3.437 | 2.584–4.571 | <0.001 |
| Headache (yes vs. no) | 0.552 | 0.371–0.821 | 0.003 |
| Anosmia (yes vs. no) | 0.563 | 0.387–0.821 | 0.003 |

AKI, acute kidney injury; CNI, calcineurin inhibitor; eGFR, estimated glomerular filtration rate; MPA, mycophenolate acid.

Footnotes: Variables included in the modeling: age, type of donor (living vs. deceased), time after transplantation; hypertension, diabetes, previous cardiovascular disease, lung disease, and neoplasia as comorbidities; CNI-MPA as the maintenance immunosuppression, recent use of high dose of steroids; time of the start of COVID-19 symptoms; expectoration, dyspnea, runny nose, headache, nasal congestion, asthenia, myalgia, nausea and vomiting, anosmia, and ageusia as COVID-19 symptoms.

Values with missing values < 10%: time after transplantation and recent use of high dose of steroids. For the time after transplantation the median value was imputed, and for the use of steroid the mode value, i.e., no use.

Variables excluded from the modeling:
  a) number of missing were higher than 10%: eGFR (10.9%) and AKI (37.4%)
  b) collinearity: hypoxemia (colinear with dyspnea).

among KT recipients, the nosocomial source is relevant because this group of patients more frequently need hospitalization than the general population, owing to complications that are inherent to the transplant itself, such as acute rejection, infections, and neoplasia. Another aspect is the unavoidable exposure to the nosocomial environment at the time of transplantation. Since the beginning of the pandemic, many programs have temporarily reduced the transplantation activity, mainly those performed using living donors [18, 19]; however, because of its nature, KT activities were not completely interrupted. Although it has not been completely measured, the risk of infection by SARS-CoV-2 in the first days after transplantation should not be overlooked, especially because infected patients in the preclinical phase, even with a negative PCR test and normal pulmonary imaging findings, can inadvertently undergo transplantation [20]. Despite this potential risk, we observed that the recipients enrolled in this cohort had a long interval between transplantation and the COVID-19 diagnosis (only 2.4% had a < 1 month interval), which is similar to that reported in other similar studies [12, 21]. Finally, most cases of nosocomial infection occurred at the beginning of the pandemic, which can suggest a probable learning curve among hospital teams that managed those patients.

Unlike other diseases caused by respiratory viruses, one particular characteristic of COVID-19 is the great variety of clinical presentations, despite the clear predominance of respiratory symptoms such as cough and dyspnea, followed by fever and other systemic symptoms [4, 5, 8]. This scenario has not been different among transplant patients, 40% to 70% of whom had cough, dyspnea, and fever [9–12, 21]. Meanwhile, gastrointestinal symptoms (mainly diarrhea) were observed in about 30% of KT recipients, which is different from the observed frequency of these symptoms in non-transplant patients (< 5%) [11, 12]; however,

the reason for this finding is unclear. Some studies have described the presence of replicant virus in the feces of infected patients; however, the direct effects of SARS-CoV-2 in the gastrointestinal tract are not well known [22, 23]. It is possible that transplant recipients are more susceptible to gastrointestinal manifestations due to co-infections by intestinal parasites or cytomegalovirus [24, 25]. Another possible explanation for this common finding is the use of immunosuppressive drugs related to gastrointestinal adverse events, such as mycophenolate, in KT recipients [26]. Interestingly, in our population diarrhea was an independent predictor of hospitalization, even though it was not associated with the risk of death.

Considering that COVID-19 is a predominantly respiratory disease with a high potential for progressing to severe acute respiratory syndrome, two clinical presentations are considered fundamental to defining the severity and to deciding the initial clinical management: dyspnea and hypoxemia. As expected, dyspnea was a predictor of hospitalization and death in our population. Severe acute respiratory syndrome has been the main cause of hospitalization [27]; however, even among transplanted patients who have a potential risk for progressing to a more severe presentation, some presented with a discrete disease characterized by coryza, rhinorrhea, cough, or minor systemic symptoms such as myalgia, ageusia, and anosmia. The initial stratification of severity has been considered one of the most relevant and challenging aspects in the clinical management of COVID-19, with a possible allocation of patients to home care [28, 29], which was reported in a third of the patients in the present cohort.

As there is no effective and specific treatment for COVID-19, the clinical management considerably varied in the first months of the pandemic, mainly depending on the patient allocation (home care or in-hospital care), as detailed in Table 2, and in the period of the pandemic. Despite the lack of evidence, antibiotics were universally used in in-hospital patients and drugs with an attributable but questionable antiviral effect, such as ivermectin, were more frequently used in patients who were treated at home. Hydroxychloroquine was indicated for some in-hospital patients, a practice that has been controversial since the beginning of the pandemic and was reported in 70% of the transplanted patients evaluated from March to May 2020 in an international multicenter study [12]. The use of hydroxychloroquine was initially adopted as a "compassionate" strategy; however, considering the emergence of evidence demonstrating its lack of efficacy [30] and a potential lack of safety [31], this practice has been abandoned by the majority of the centers in the last months. In fact, the basis of clinical management was adjustment in immunosuppressive medications and the use of high-dose steroids, especially after the RECOVERY study [32], despite the lack of evidence for this specific population. In respect to immunosuppression, we presented only the immunosuppressive regimen that patients were using at the COVID-19 diagnosis, and the initial regime was not available. Of note, the time interval between transplantation and COVID-19 infection was 5.9 years, however we consider that the lack of this information should be highlight as a limitation.

The reduction or interruption of immunosuppressives in KT recipients in the context of infection must be considered; however, the benefit of infection control should be balanced with the risk of acute rejection. In the present study, there was a need for adjustment in the immunosuppressive schedule in about 70% of the in-hospital patients, similar to that reported in other studies [11, 12, 21]. We did not explore the specific reasons for the choice in each patient; however, generally, the option was to reduce the dose of antimetabolic agents or their withdrawal. The preferential management with respect to antimetabolic drugs followed the recommendations based on the opinion of specialists [33, 34] and those reported in other observational studies [12, 35]. Since the first reports, lymphopenia has been described as a potential predictor of worse prognosis [4, 8]; thus, despite the lack of clear evidence, this strategy can be explained by the impact of COVID-19 on the counts of white blood cells, mainly lymphocytes, which can be amplified or sustained by this class of drugs [11].

The fatality rate among our patients was 21%, whereas that in the general Brazilian population was 2.5%, as reported by the Health Ministry [1]. This line of comparison is merely speculative, considering that the sources of information and methods are completely distinct. However, by analyzing the fatality rate grouped by intermediate outcomes, it is possible to design better-quality scenarios for comparison. For instance, the rates of deaths among KT recipients who required hospitalization have varied from 26% to 32% in previous studies [12, 21], comparable to the rate of 31.6% observed in the present study. We observed a 58.2% fatality rate among patients who were admitted to the ICU and 75.7% among those who required MV. Surprisingly, these rates were not higher than those observed among the first 250,0000 Brazilians diagnosed with COVID-19 who received in-hospital clinical management between February and August 2020. However, it is important to highlight that the general population was older: 60 vs. 51 years old on average [6]. Similar to what was observed in our cohort, the global fatality rate was 38% in a previous population study, being 59% among patients admitted to the ICU and 80% in those who required MV [6]. Nevertheless, it is important to note that large regional disparities were observed, with rates of deaths among Brazilians aged < 60 years of 31% and 15% in the northeast and south regions, respectively [6]. In addition, some transplant recipients in Brazil have easier access to secondary and tertiary health-care centers, which can be related to earlier evaluation and better clinical management.

It is natural to speculate about the preponderant importance of prolonged and unavailable use of immunosuppressive drugs as a determinant factor of prognosis. However, it is possible that the presence of comorbidities had a stronger prognostic impact than immunosuppression would have per se. As expected, there was a high prevalence of hypertension, diabetes, and previous cardiovascular disease in the patients enrolled in the present study, which would explain the high risk for worse outcomes. Furthermore, using propensity scores, a French study did not find a higher risk of mortality when KT recipients were matched for age, sex, body mass index, diabetes, cardiovascular disease, hypertension, chronic pulmonary disease, and baseline kidney function with non-transplanted patients, suggesting that the effect of immunosuppression is less important than comorbidities [21].

Finally, to date, one factor that has been described as a predictor of death in patients diagnosed with COVID-19 is AKI, mainly when RRT is required [36]. In the first reports on citizens in the New York area, AKI was present in 22.2% of patients who needed hospitalization; however, the incidence in patients who did not survive was 72.1% [8]. The risk of AKI is significantly associated with age, diabetes, and previous cardiovascular disease, besides MV or vasopressor requirement [36]. In our study, the 90-day cumulative incidence of death was significantly higher in patients who had AKI at the COVID-19 diagnosis and in those who required RRT during hospitalization. AKI in KT recipients can be considered a two-way clinical situation: it is a marker of severity in the course of the infection, especially in patients with excess comorbidities, but it adds risks including impairment in ventilation management due to volume overload, risk of secondary infection (such as those associated with use of intravascular dispositive), and risk of bleeding. It is important to emphasize that information about kidney biopsy after the COVID-19 diagnosis was not available in the present study, thus this lack of information is a limitation to understand the results about acute kidney injury.

Despite being the largest cohort study that enrolled KT recipients with COVID-19 to date, this study had several limitations that should be considered, with its retrospective design being one of the most relevant. As inherent to any retrospective study, there was missing information, although its number was low considering the total number of patients included; some definition could not be standardized, such as the classification of the source of infection; and it is possible that some patients with mild symptoms were not diagnosed, which can cause a potential selection bias. Finally, considering the observational nature of the study, the clinical

management could not be standardized, mainly those related to advanced supportive care, such as the indication for ICU admission and MV. Besides, Brazil is a continental country, which suffers with deep social inequality, but unfortunately information on socioeconomic status were not available. On the other hand, it is important to emphasize that this study provides real-life evidence, with some degree of strength owing to the large number of included patients from multiple centers accounting for the majority of transplantation activities in Brazil. In addition, Brazil has the largest public transplantation system worldwide, with the second largest number of transplanted patients being followed up during the time when it was one of the epicenters of the COVID-19 pandemic.

In conclusion, KT recipients diagnosed with COVID-19 had a long interval between the time of transplantation and the time of the diagnosis, and most of them had multiple comorbidities. As expected, respiratory symptoms were frequent; however, the presence of diarrhea was more common in KT recipients than in the general population. An important point for in-hospital patients was the management of immunosuppressive medications, although there is no robust evidence suggesting the best approach. One in every five patients died, and the rate of death was significantly higher in those who had AKI, mainly when RRT was required.

## Supporting information

**S1 Table. Institutional Review Boards (IRB).** Footnote: In all cases where informed consent was obtained, it was written. In some specific cases, the informed consent was waived by the local ethics committees. Legend: IRB, Institutional Review Board; PI, principal investigator.
(DOCX)

**S2 Table. COVID-19-attributable symptoms codebook.** Footnote: the attributable symptoms were defined by the local investigator.
(DOCX)

**S3 Table. Outcomes according to the time after transplantation.** Legend: AKI, acute kidney injury; ICU, intensive care unit; MV, mechanical ventilation; RRT, renal replacement therapy.
(DOCX)

**S4 Table. Comparison between hospitalized and non-hospitalized patients.** Footnote: Missing values: BMI = 92 (5.5%); time after transplantation = 4 (0.24%); eGFR = 183 (10.9%); Recent high dose of steroids = 33 (2.0%); Recent use of Thymoglobulin = 50 (3.0%); AKI = 628 (37.4%). Legend: AKI, acute kidney injury; AZA, azathioprine; BMI, body mass index; CNI, calcineurin inhibitor; DD, deceased donor; eGFR, estimated glomerular filtration rate; IS, immunosuppressive; KAL, kidney after liver; KT, kidney transplant; LD, living donor; mTORi, mammalian target of rapamycin inhibitor; PAK, pancreas after kidney; SHK, simultaneous heart-kidney; SLK, simultaneous liver-kidney; SPK, simultaneous pancreas-kidney.
(DOCX)

**S5 Table. Comparison between survivors and non-survivors.** Footnotes: Missing values: BMI = 92 (5.5%); time after transplantation = 4 (0.24%); eGFR = 183 (10.9%); Recent high dose of steroids = 33 (2.0%); Recent use of Thymoglobulin = 50 (3.0%); AKI = 628 (37.4%). Legend: AKI, acute kidney injury; AZA, azathioprine; BMI, body mass index; CNI, calcineurin inhibitor; DD, deceased donor; eGFR, estimated glomerular filtration rate; IS, immunosuppressive; KAL, kidney after liver; KT, kidney transplant; LD, living donor; mTORi, mammalian target of rapamycin inhibitor; PAK, pancreas after kidney; SHK, simultaneous heart-kidney; SLK, simultaneous liver-kidney; SPK, simultaneous pancreas-kidney.
(DOCX)

**S1 Data.**
(XLSX)

## Acknowledgments

The authors thank Associação Brasileira de Transplantes de Órgãos (ABTO) for all the support received, Mônica Rika Nakamura for her assistance in the regulatory process and the Gerência de Ensino e Pesquisa (GEP) do Complexo Hospitalar da Universidade Federal do Ceará (CH-UFC), notedly Antonio Brazil Viana Junior, for enabling the use of REDcap.

## The *COVID-19-KT Brazil Study* includes the following participants:

Gisele Meinerz, MD, PhD (Santa Casa de Misericórdia de Porto Alegre, Porto Alegre, RS, Brazil), Aline Lima Cunha Alcântara, MD, MSc (Hospital Universitário Walter Cantídio, Fortaleza, CE, Brazil), Celi Melo Girão, RN, MSc (Hospital Geral de Fortaleza, Fortaleza, CE, Brazil), Ida Maria Maximina Fernandes Charpiot, MD, PhD (Hospital de Base de SJRP, SJRP, SP, Brazil), Teresa Cristina Alves Ferreira, MD, PhD (Hospital Universitário da UFMA, São Luis, MA, Brazil), Rodrigo Fontanive Franco, MD (Hospital de Clínicas de Porto Alegre, Porto Alegre, RS, Brazil), Tomás Pereira Júnior, MD (Hospital Universitário Onofre Lopes, Natal, RN, Brazil), Maria Eduarda Heinzen de Almeida Coelho, MD (Hospital Santa Isabel, Blumenau, SC, Brazil), Marilda Mazzali, MD, PhD (University of Campinas–UNICAP, Scholl of Medical Sciences, Division of Nephrology, Renal Transplant Unit, Renal Transplant Research Laboratoy, Campinas, SP, Brazil), Gustavo Fernandes Ferreira, MD, PhD (Santa Casa de Misericórdia de Juiz de Fora, Juiz de Fora, MG, Brazil), Viviane Brandão Bandeira de Mello Santana, MD (Hospital de Base de Brasília, Brasilia, DF, Brazil), Nicole Gomes Campos Rocha, MD (Hospital de Base de Brasília, Brasilia, DF, Brazil), Anita Leme da Rocha Saldanha, MD (Hospital Beneficência Portuguesa, São Paulo, SP, Brazil), Tania Leme da Rocha Martinez, MD, PhD (Hospital Beneficência Portuguesa, São Paulo, SP, Brazil), Joao Egidio Romao Junior, MD, PhD (Hospital Beneficência Portuguesa, São Paulo, SP, Brazil), Maria Regina Teixeira Araújo, MD, PhD (Hospital Beneficência Portuguesa, São Paulo, SP, Brazil), Irene de Lourdes Noronha, MD, PhD (Hospital Beneficência Portuguesa, São Paulo, SP, Brazil), Sibele Lessa Braga, MD (Hospital Beneficência Portuguesa, São Paulo, SP, Brazil), Marina Abritta Hanauer, MD (Hospital Municipal São José de Joinville, Fundação Pró-Rim, Joinville, SC, Brazil), Elen Almeida Romão, MD, PhD (Hospital das Clinicas de Ribeirão Preto, Ribeirão Preto, SP, Brazil), Auro Buffani Claudino, MD, MSc (Hospital Santa Marcelina, São Paulo, SP, Brazil), Gustavo Guilherme Queiroz Arimatea, MD, MSc (Hospital Universitário da UnB, Brasília, DF, Brazil), Lívia Cláudio de Oliveira, MD (Hospital Universitário da UnB, Brasília, DF, Brazil), Deise De Boni Monteiro de Carvalho, MD, (Hospital São Francisco na Providência de Deus, Rio de Janeiro, RJ, Brazil), Tereza Azevedo Matuck, MD (Hospital São Francisco na Providência de Deus, Rio de Janeiro, RJ, Brazil), Alexandre Tortoza Bignelli, MD, MSc (Hospital Universitário Cajuru, Curitiba, PR, Brazil), Silvia Regina Hokazono, MD (Hospital Universitário Cajuru, Curitiba, PR, Brazil), José Hermógenes Rocco Suassuna, MD, PhD (Hospital Universitário Pedro Ernesto, Rio de Janeiro, RJ, Brazil), Suzimar da Silveira Rioja, MD, PhD (Hospital Universitário Pedro Ernesto, Rio de Janeiro, RJ, Brazil), Rafael Lage Madeira, MD (Hospital Felicio Rocho, Belo Horizonte, MG, Brazil), Sandra Simone Vilaça, MD (Hospital Felicio Rocho, Belo Horizonte, MG, Brazil), Sandra Simone Vilaça, MD (Hospital Felicio Rocho, Belo Horizonte, MG, Brazil, Carlos Alberto Chalabi Calazans, MD, PhD (Hospital Marcio Cunha, Ipatinga, MG, Brazil), Daniel Costa Chalabi Calazans, MD (Hospital Marcio Cunha, Ipatinga, MG, Brazil), Luiz Antonio Miorin, MD, PhD (Santa Casa de Misericórdia de

São Paulo, São Paulo, SP, Brazil), Patricia Malafronte, MD, PhD (Santa Casa de Misericórdia de São Paulo, São Paulo, SP, Brazil), Larissa Guedes da Fonte Andrade, MD, MSc (Universidade Federal de Pernambuco, Recife, PE, Brazil), Filipe Carrilho de Aguiar, MD, MSc (Universidade Federal de Pernambuco, Recife, PE, Brazil), Fabiana Loss de Carvalho, MD (Hospital do Rocio, Curitiba, PR, Brazil), Karoline Sesiuk Martins, MD (Hospital do Rocio, Curitiba, PR, Brazil), Hélady Sanders Pinheiro, MD, PhD (Hospital Universitário da UFJF, Juiz de Fora, MG, Brazil), Emiliana Spadarotto Sertório, MD (Hospital Universitário da UFJF, Juiz de Fora, MG, Brazil), André Barreto Pereira, MD, PhD (Hospital Marieta Konder Bornhausen, Itajai, SC, Brazil), David José Barros Machado, MD, PhD (Hospital Alemão Osvaldo Cruz, Sao Paulo, SP, Brazil), Carolina Maria Pozzi, MD (Hospital Universitário Evangélico de Curitiba, Curitiba, PR, Brazil), Leonardo Viliano Kroth MD, PhD (Hospital São Lucas, Porto Alegre, RS, Brazil), Lauro Monteiro Vasconcellos Filho, MD, PhD (Hospital Meridional, Cariacica, ES, Brazil), Rafael Fabio Maciel, MD, MSc (Hospital Nossa Senhora das Neves, João Pessoa, PB, Brazil), Amanda Maíra Damasceno Silva, MD (Hospital Antônio Targino, Campina Grande, PB, Brazil), Ana Paula Maia Baptista, MD, MsC (Hospital São Rafael, Salvador, BA, Brazil), Pedro Augusto Macedo de Souza, MD (Santa Casa de Belo Horizonte, Belo Horizonte, MG, Brazil), Marcus Faria Lasmar, MD, PhD (Hospital Universitário Ciências Médicas, Belo horizonte, MG, Brazil), Luciana Tanajura Santamaria Saber, MD, PhD (Santa Casa de Ribeirão Preto, Ribeirão Preto, SP, Brazil), Lilian Monteiro Pereira Palma, MD, PhD (Centro Médico de Campinas, Campinas, SP, Brazil)

## Author Contributions

**Conceptualization:** Lúcio R. Requião-Moura, Tainá Veras de Sandes-Freitas, Laila Almeida Viana, Marina Pontello Cristelli, Luis Gustavo Modelli de Andrade, Helio Tedesco-Silva, José Medina-Pestana.

**Data curation:** Lúcio R. Requião-Moura, Tainá Veras de Sandes-Freitas, Laila Almeida Viana, Marina Pontello Cristelli, Luis Gustavo Modelli de Andrade, José Medina-Pestana.

**Formal analysis:** Lúcio R. Requião-Moura, Tainá Veras de Sandes-Freitas, Laila Almeida Viana, Marina Pontello Cristelli, Luis Gustavo Modelli de Andrade, Helio Tedesco-Silva.

**Funding acquisition:** Tainá Veras de Sandes-Freitas.

**Investigation:** Lúcio R. Requião-Moura, Tainá Veras de Sandes-Freitas, Marina Pontello Cristelli, Luis Gustavo Modelli de Andrade, Valter Duro Garcia, Claudia Maria Costa de Oliveira, Ronaldo de Matos Esmeraldo, Mario Abbud Filho, Alvaro Pacheco-Silva, Katia Cronemberger Sousa, Alessandra Rosa Vicari, Kellen Micheline Alves Henrique Costa, Denise Rodrigues Simão, Marcos Vinicius de Sousa, Juliana Bastos Campos, Ricardo Augusto Monteiro de Barros Almeida, Luciane Mônica Deboni, Miguel Moysés Neto, Juliana Aparecida Zanocco.

**Methodology:** Lúcio R. Requião-Moura, Tainá Veras de Sandes-Freitas.

**Project administration:** Lúcio R. Requião-Moura, Tainá Veras de Sandes-Freitas.

**Supervision:** Lúcio R. Requião-Moura, Tainá Veras de Sandes-Freitas.

**Validation:** Lúcio R. Requião-Moura, Tainá Veras de Sandes-Freitas.

**Visualization:** Tainá Veras de Sandes-Freitas.

**Writing – original draft:** Lúcio R. Requião-Moura, Tainá Veras de Sandes-Freitas, Helio Tedesco-Silva.

**Writing – review & editing:** Lúcio R. Requião-Moura, Tainá Veras de Sandes-Freitas, Valter Duro Garcia, Claudia Maria Costa de Oliveira, Ronaldo de Matos Esmeraldo, Mario Abbud Filho, Alvaro Pacheco-Silva, Katia Cronemberger Sousa, Alessandra Rosa Vicari, Kellen Micheline Alves Henrique Costa, Denise Rodrigues Simão, Marcos Vinicius de Sousa, Juliana Bastos Campos, Ricardo Augusto Monteiro de Barros Almeida, Luciane Mônica Deboni, Miguel Moysés Neto, Juliana Aparecida Zanocco, Helio Tedesco-Silva, José Medina-Pestana.

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
