## [Decision Letter · Decision Letter 0]

14 Apr 2021

PONE-D-21-05548

High mortality among kidney transplant recipients diagnosed with coronavirus disease 2019: Results from the Brazilian multicenter cohort study.

PLOS ONE

Dear Dr. Requião-Moura,

Thank you for submitting your manuscript to PLOS ONE. After careful consideration, we feel that it has merit but does not fully meet PLOS ONE’s publication criteria as it currently stands. Therefore, we invite you to submit a revised version of the manuscript that addresses the points raised during the review process.

**The manuscript focuses on a topic of current interest. The study, however, presents several shortcomings that the authors need to address. To mention some of them, i) unclear when the informed consent was obtained from all patients; ii) unclear the criteria for diagnosis of COVID-19 infection; iii) need to elaborate the means by which the symptoms were recorded on the REDcap system by the transplant/primary care physicians; iv) need to clarify when the serology test was done in the 15.3% of patients who were diagnosed by means of serology rather than in a PCR; v) need to perform multivariate analysis of the outcomes including age and frailty; vi) unclear what percent of the transplant population in Brazil is Afro-Brazilian; vii) unclear whether they have data on transplant induction therapy (lymphocyte depleting versus non-depleting or no induction); viii) need to provide the time between symptoms and hospitalization in those hospitalized; ix) unclear whether nosocomial infections occur at the time of transplantation or while admitted for another cause; x) useful to have information on socioeconomic status in the current study; xi) need to provide the outcome in patients transplanted early (<1 year) vs late (>1 year); xii) need to perform a comparison between hospitalized and non-hospitalized, as well as survivors vs non-survivors; xiii) unclear whether any patients have a kidney biopsy and if there were any episodes of rejection.**

We look forward to receiving your revised manuscript.

Kind regards,

Giuseppe Remuzzi

Academic Editor

PLOS ONE

Journal Requirements:

Please include a list of participating research centres, and the name of the ethics committee at each that approved this study in the supporting information."

3. We note that you obtained consents from participants to take part in your retrospective study. In the Ethics Statement on the online submission form and the manuscript Methods, please clarify the context in which consent was obtained, and specify whether patients provided:

                1) Consent to use their medical records/samples in research

                2) Consent to undergo a procedure

                3) Consent to take part in the study reported in this manuscript.

If the ethics committee waived the need for additional informed consent, please state this. Please also ensure that you have specified the type of consent contained (written or verbal, and if verbal, how consent was documented and witnessed.)

Thank you for your attention to these requests.

4, Thank you for stating in your Funding Statement:

This study was partially supported by Novartis Pharma Brazil

5a) If there are ethical or legal restrictions on sharing a de-identified data set, please explain them in detail (e.g., data contain potentially identifying or sensitive patient information) and who has imposed them (e.g., an ethics committee). Please also provide contact information for a data access committee, ethics committee, or other institutional body to which data requests may be sent.

5b) If there are no restrictions, please upload the minimal anonymized data set necessary to replicate your study findings as either Supporting Information files or to a stable, public repository and provide us with the relevant URLs, DOIs, or accession numbers. Please see http://www.bmj.com/content/340/bmj.c181.long for guidelines on how to de-identify and prepare clinical data for publication. For a list of acceptable repositories, please see http://journals.plos.org/plosone/s/data-availability#loc-recommended-repositories.

6, One of the noted authors is a group or consortium COVID-19-KT Brazil. In addition to naming the author group, please list the individual authors and affiliations within this group in the acknowledgments section of your manuscript. Please also indicate clearly a lead author for this group along with a contact email address.

Reviewers' comments:

Reviewer's Responses to Questions

**Comments to the Author**

1. Is the manuscript technically sound, and do the data support the conclusions?

Reviewer #1: Yes

Reviewer #2: Yes

2. Has the statistical analysis been performed appropriately and rigorously? 

Reviewer #1: Yes

Reviewer #2: Yes

3. Have the authors made all data underlying the findings in their manuscript fully available?

Reviewer #1: Yes

Reviewer #2: Yes

4. Is the manuscript presented in an intelligible fashion and written in standard English?

Reviewer #1: Yes

Reviewer #2: Yes

5. Review Comments to the Author

Reviewer #1: The study on mortality in kidney transplant recipients with coronavirus disease from the Brazilian transplant centres is a large cohort analysed for the effects of COVID-19 infection in kidney transplant recipients. The study shows our mortality of 21% would you creases with hospital admission and ICU admission to 31 and 76% respectively.

The study is well conducted with a large patient population from 35 Centres in Bazil.

The informed consent was obtained from all patients but perhaps it’s not clear when the content was obtained as the study was a retrospective analysis.

The criteria for diagnosis of COVID-19 infection is not very clear. The study mentioned diagnosis was according to the clinical criteria for each centre. However the definition could’ve been different in the different centres which requires a mention in the article.

The authors should be congratulated for collecting data on symptoms. However the authors could elaborate the means by which the symptoms were recorded on the REDcap system by the transplant/primary care physicians.

The author should also clarify when was the serology test done in the 15.3% of the patients who were diagnosed by means of serology rather than are in a PCR.

The authors have emphasised the role of acute kidney injury and RRT on outcome. A multivariate analysis of the outcomes including age and frailty possible would be very useful.

Reviewer #2: The authors of the manuscript “High mortality among kidney transplant recipients diagnosed with coronavirus disease 2019: Results from the Brazilian multicenter cohort study” should be commended for performing the largest cohort study of COVID19 in kidney transplant recipients published to date. Besides being the largest, the authors were able to capture migratory patterns of infection within Brazil and nicely explore the continuum of infection from diagnosis to 90 days. In addition, informed consent was obtained. This allows insight into both outpatient and inpatient mortality rates among other insights, a feature lacking in several other published manuscripts on the subject. Although I am happy with the manuscript as is, I wonder if the authors could explore a few other questions of interest, as their large database may allow further analysis. See questions below:

1. I notice only 11.4% of patients were Afro-Brazilian. What percent of the transplant population in Brazil is Afro-Brazilian?

2. Do the authors have data on transplant induction therapy? Lymphocyte depleting versus non-depleting or no induction.

3. What was the time between symptoms and hospitalization in those hospitalized? This data would be useful for clinicians to know when patients are at the highest risk for hospitalization.

4. Nosocomial infections were high. Did these occur at the time of transplantation or while admitted for another cause?

5. The authors mention deep social inequality in Brazil. Do they have any information on socioeconomic status in the current study?

6. As the authors have a good number of patients perhaps they analyze the following:

a. Outcomes in patients transplanted early (<1 year) vs late (>1 year)

b. Outcomes in patients without patient reduction in immunosuppression versus no change. It would be interesting to know if early reduction in immunosuppression can help avoid hospital admission or perhaps it makes no difference.

c. Can the authors perform a comparison between hospitalized and non-hospitalized, as well as survivors vs. non-survivors. Were there any other characteristics besides AKI that predicted a higher risk for adverse outcome (for example age, race, co-morbid conditions, regions, or times of disease)?

d. Did any patients have a kidney biopsy? Were there any episodes of rejection?

6. PLOS authors have the option to publish the peer review history of their article (what does this mean?). If published, this will include your full peer review and any attached files.

Reviewer #1: No

Reviewer #2: No

---

## [Author Response · Author response to Decision Letter 0]

11 Jun 2021

Dear Academic Editor

Dr. Giuseppe Remuzzi

On behalf of authors, we thank the opportunity to resubmit to PLOS ONE the revised version of our manuscript entitled “High mortality among kidney transplant recipients diagnosed with coronavirus disease 2019: Results from the Brazilian multicenter cohort study.” (PONE-D- 21-05548).

Your comments as well as the comments of the reviewers were crucial to improve our manuscript and all issues raised by you are following addressed.

Lúcio R. Requião-Moura

Review 

Academic editor

The manuscript focuses on a topic of current interest. The study, however, presents several shortcomings that the authors need to address. To mention some of them:

From the editor:

i) unclear when the informed consent was obtained from all patients

Author comments:

Thank you for the opportunity to clarify this statement.

The informed consent was obtained to have access to the patients’ medical records. As the study was a retrospective analysis, in cases of exemption, it followed specific national legislations, the local Institutional Review Board (IRB) recommendations, and the guidelines of the Declaration of Helsinki. 

It was clarified in the methods section, as following:

“Informed consent or its exemption followed specific national legislations, the local Institutional Review Board (IRB) recommendations, and the guidelines of the Declaration of Helsinki.”

From the editor:

ii) unclear the criteria for diagnosis of COVID-19 infection

Author comments:

The COVID-19 diagnosis was considered if patients presented at least one COVID-19-attributable symptom associated with a positive test. A codebook with the COVID-19-attributable symptoms was included in the supplementary material.

The tests to confirm the COVID-19 diagnosed were (one of them, at least): polymerase chain reaction, serology, or viral antigen detection. The diagnosis based on serology was retrospectively performed in patients who had had attributable symptoms (4 weeks after the beginning of the symptoms, in average).

It was clarified in the methods section, as following:

“The diagnosis was considered only in patients who presented at least one COVID-19-attributable symptom associated with a positive result in any one of the following tests: polymerase chain reaction (PCR), serology, or viral antigen detection. A codebook with the description of the attributable symptoms is presented in the supplementary table 1. The diagnosis based on serology was retrospectively performed in patients who had had attributable symptoms. The attributable symptoms were defined by the local investigator"

From the editor:

iii) need to elaborate the means by which the symptoms were recorded on the REDcap system by the transplant/primary care physicians.

Author comments:

A codebook with the COVID-19-attributable symptoms was included in the supplementary material (supplementary table 1). The investigators could check one or more symptoms in the list using a check box. Symptoms or signs not listed in the codebook could be reported in a descriptive field.

From the editor:

iv) need to clarify when the serology test was done in the 15.3% of patients who were diagnosed by means of serology rather than in a PCR

Author comments:

The diagnosis based on serology was retrospectively performed in patients who had had attributable symptoms. The time between the beginning of symptoms and the serology was 26.0 (9.0; 55.0) days.

In the current version of the manuscript, we clarified this information in the methods sections and the time between the beginning of symptoms and serology was included in the results.

In the methods:

“The diagnosis based on serology was retrospectively performed in patients who had had attributable symptoms”.

In the results:

“Among patients who had the diagnosis based on serology, the time between the COVID-19 attributable symptoms beginning and serology was 26.0 (9.0; 55.0) days”.

From the editor:

v) need to perform multivariate analysis of the outcomes including age and frailty.

Author comments:

Thank you for this suggestion.

We performed multivariate analysis for the probability of hospitalization and death (comments in the topic xii). Unfortunately, information about frailty was not available. 

The results of the multivariable analysis were depicted in two tables in the current version of the manuscript: 4 and 5.

From the editor:

vi) unclear what percent of the transplant population in Brazil is Afro-Brazilian.

Author comments:

There are no official data about race among transplanted patients in Brazil. Moreover, Brazil is a multiracial country, and the definition of race is self-reported, what has more correlation with the social identity than a biological background. 

In a recent multicenter study (n=3,992) carried out to investigate the incidence of delayed graft function after kidney transplant performed with deceased donors the frequency of Afro-Brazilian was 13.7% (DOI: doi:10.1111/tri.13865). 

Additionally, a previous study that evaluated 10,000 kidney transplants performed in a Brazilian single center, the frequency of Afro-Brazilian was 23% (DOI: 10.1007/s40620-017-0379-9). 

These frequencies are similar those that was observed in the present study.

From the editor:

vii) unclear whether they have data on transplant induction therapy (lymphocyte depleting versus non-depleting or no induction).

Author comments:

Data on transplant induction therapy were not available.

Actually, we presented the immunosuppressive regimen that patients were using at the COVID-19 diagnosis. Of note, the time interval between transplantation and COVID-19 infection was 5.9 years, however we consider that the lack of this information could be high light as a limitation. We included a comment about this limitation in the discussion of the current version of the manuscript.

In respect to immunosuppression, we were especially interested in the use of high doses of steroids or lymphocyte depleting antibody to treat acute rejection until 3 months from the COVID-19 diagnosis, considered as recent use. 

Recent use of high dose of steroids was recorded for 4.3% (n=73) patients, whereas lymphocyte depleting antibody was recorded for 2.8% (n=47).

These results were embodied in the results, as following:

“Use of high dose of steroids and lymphocyte depleting antibody to treat acute rejection until 3 months before the COVID-19 diagnosis were recorded for 4.3% (n=73) and 2.8% (n=47), respectively.”

From the editor:

viii) need to provide the time between symptoms and hospitalization in those hospitalized.

Author comments:

The time between symptoms and hospitalization was 7.0 (4.0; 10.0) days. This information was included in the results of the current version of the manuscript, as following:

“About two-thirds of the patients (65.1%) were hospitalized for clinical management, 7.0 (4.0; 10.0) days after the beginning of symptoms.”

From the editor:

ix) unclear whether nosocomial infections occur at the time of transplantation or while admitted for another cause.

Author comments:

Thank you for the opportunity to clarify this information: 137 patients had the nosocomial infection as the source of infection. Among them, in 16 (11.7%) the infection occurred during the hospitalization when the transplantation was performed. We embodied this information in the current version of the manuscript.

From the editor:

x) useful to have information on socioeconomic status in the current study.

Author comments:

The socioeconomic status was not available in the present study. We agree that it is a very important point, and it should be addressed in future investigations. We are including this statement in the discussion.

From the editor:

xi) need to provide the outcome in patients transplanted early (<1 year) vs late (>1 year).

Author comments:

Thank you for this suggestion.

The outcomes were compared according to the time of transplantation, which was dichotomized in ≤1 year and >1 years. These results were included in the supplementary table 2 in the current version of the manuscript.

The hospitalization rate was higher among patients who had the time after transplantation shorter than 1 year (71.3% vs. 64.0%, P=0.031), as well as the requirement for RRT (30.6% vs. 22.1, P=0.005).

The following sentence was included in the results in the current version:

“The outcomes were compared according to the time of transplantation: longer or shorter than 1 year (supplementary table 2). The hospitalization rate was higher among patients who had the time after transplantation shorter than 1 year (71.3% vs. 64.0%, P=0.031), as well as the requirement for RRT (30.6% vs. 22.1, P=0.005).”

From the editor:

xii) need to perform a comparison between hospitalized and non-hospitalized, as well as survivors vs non-survivors.

Author comments:

Thank you for this suggestion.

We compared the variables according to the hospitalization (yes vs. no) and death (survivors vs. non-survivors). The continuous variables were compared by Mann-Whitney U test and the categorical were compared by X2 or Fisher test. We are including these results in the current version of the manuscript. 

In the supplementary table 3 is shown the comparison between patients who were hospitalized versus those who were not. The variables that reached a P-value ≤ 0.10 in the univariable comparison were selected for the multivariable modeling. The multivariable analysis was performed by the binary logistic regression. The final results are shown in the table 4. 

In summary, the variables related with the probability of hospitalization were age, hypertension, previous cardiovascular disease, recent use of high dose of steroid, and fever, dyspnea, diarrhea, and nausea/vomiting as COVID-19 symptoms. On the other hand, the variables that reduced the probability of hospitalization were time of COVID-19 symptoms, and nasal congestion, headache, arthralgia and anosmia as COVID-19 symptoms.

In the supplementary table 4 is shown the comparison between patients who survived versus those who did not. Similarly, the variables that reached a P-value ≤ 0.10 in the univariable comparison were selected for the multivariable modeling. The multivariable analysis was performed by the binary logistic regression. The final results are shown in the table 5.

The variables related with the probability of death within 90 days after COVID-19 were age, time after transplantation, presence of hypertension, previous cardiovascular disease, use of tacrolimus and mycophenolate as the maintenance immunosuppression, recent use of high dose of steroids, and dyspnea as COVID-19 symptom. On the other hand, the variables that reduced the risk of death were time of symptoms, and headache and anosmia as COVID-19 symptoms. 

From the editor:

xiii) unclear whether any patients have a kidney biopsy and if there were any episodes of rejection.

Author comments:

Unfortunately, information about kidney biopsy after the COVID-19 diagnosis was not available. We consider that this lack of information is a limitation to understand the results about acute kidney injury, and we included this limitation in the discussion. 

Reviewer #1 comments:

The informed consent was obtained from all patients but perhaps it’s not clear when the content was obtained as the study was a retrospective analysis

Author comments:

Thank you for the opportunity to clarify this statement.

The informed consent was obtained to have access to the patients’ medical records. As the study was a retrospective analysis, in cases of exemption, it followed specific national legislations, the local Institutional Review Board (IRB) recommendations, and the guidelines of the Declaration of Helsinki. 

It was clarified in the methods section, as following:

“Informed consent or its exemption followed specific national legislations, the local Institutional Review Board (IRB) recommendations, and the guidelines of the Declaration of Helsinki.”

Reviewer #1 comments:

The criteria for diagnosis of COVID-19 infection is not very clear. The study mentioned diagnosis was according to the clinical criteria for each centre. However the definition could’ve been different in the different centres which requires a mention in the article.

Author comments:

Thank you for this comment. 

The criteria for diagnosis was standardized for all centers, and it was based on two aspects: the test and COVID-19-attributable symptoms. For all patients, the COVID-19 diagnosis was considered if patients presented at least one COVID-19-attributable symptom associated with a positive test. A codebook with the COVID-19-attributable symptoms was included in the supplementary material to clarify how the information was collected.

The tests to confirm the COVID-19 diagnosed were (one of them, at least): polymerase chain reaction, serology, or viral antigen detection. The diagnosis based on serology was retrospectively performed in patients who had had attributable symptoms (4 weeks after the beginning of the symptoms, in average).

It was clarified in the methods section, as following:

“The diagnosis was considered only in patients who presented at least one COVID-19-attributable symptom associated with a positive result in any one of the following tests: polymerase chain reaction (PCR), serology, or viral antigen detection. A codebook with the description of the attributable symptoms is presented in the supplementary table 1. The diagnosis based on serology was retrospectively performed in patients who had had attributable symptoms. The attributable symptoms were defined by the local investigator”

Reviewer #1 comments:

The authors should be congratulated for collecting data on symptoms. However the authors could elaborate the means by which the symptoms were recorded on the REDcap system by the transplant/primary care physicians.

Author comments:

A codebook with the COVID-19-attributable symptoms was included in the supplementary material (supplementary table 1). The investigators could check one or more symptoms in the list using a check box. Symptoms or signs not listed in the codebook could be reported in a descriptive field.

Reviewer #1 comments:

The author should also clarify when was the serology test done in the 15.3% of the patients who were diagnosed by means of serology rather than are in a PCR.

Author comments:

It is a very important point. Thank you for your comment.

The diagnosis based on serology was retrospectively performed in patients who had had attributable symptoms. The time between the beginning of symptoms and the serology was 26.0 (9.0; 55.0) days.

In the current version of the manuscript, we clarified this information in the methods sections and the time between the beginning of symptoms and serology was included in the results.

In the methods:

“The diagnosis based on serology was retrospectively performed in patients who had had attributable symptoms”.

In the results:

“Among patients who had the diagnosis based on serology, the time between the COVID-19 attributable symptoms beginning and serology was 26.0 (9.0; 55.0) days”.

Reviewer #1 comments:

The authors have emphasised the role of acute kidney injury and RRT on outcome. A multivariate analysis of the outcomes including age and frailty possible would be very useful.

Author comments:

Thank you for your suggestion.

We performed multivariate analysis for the probability of hospitalization and death, as it was suggested by the Academic editor. Unfortunately, information about frailty was not available. 

The results of the multivariable analysis were depicted in two tables in the current version of the manuscript: 4 and 5.

In summary, the variables related with the probability of hospitalization were age, hypertension, previous cardiovascular disease, recent use of high dose of steroid, and fever, dyspnea, diarrhea, and nausea/vomiting as COVID-19 symptoms. On the other hand, the variables that reduced the probability of hospitalization were time of COVID-19 symptoms, and nasal congestion, headache, arthralgia and anosmia as COVID-19 symptoms.

Additionally, the variables related with the probability of death within 90 days after COVID-19 were age, time after transplantation, presence of hypertension, previous cardiovascular disease, use of tacrolimus and mycophenolate as the maintenance immunosuppression, recent use of high dose of steroids, and dyspnea as COVID-19 symptom. On the other hand, the variables that reduced the risk of death were time of symptoms, and headache and anosmia as COVID-19 symptoms. 

Reviewer #2 comments:

I notice only 11.4% of patients were Afro-Brazilian. What percent of the transplant population in Brazil is Afro-Brazilian?

Author comments:

In a recent multicenter study (n=3,992) carried out to investigate the incidence of delayed graft function after kidney transplant performed with deceased donors the frequency of Afro-Brazilian was 13.7% (DOI: doi:10.1111/tri.13865). 

Additionally, a previous study that evaluated 10,000 kidney transplants performed in a Brazilian single center, the frequency of Afro-Brazilian was 23% (DOI: 10.1007/s40620-017-0379-9). 

These frequencies are similar those that was observed in the present study

Reviewer #2 comments:

Do the authors have data on transplant induction therapy? Lymphocyte depleting versus non-depleting or no induction.

Author comments:

Data on transplant induction therapy were not available.

Actually, we presented the immunosuppressive regimen that patients were using at the COVID-19 diagnosis. Of note, the time interval between transplantation and COVID-19 infection was 5.9 years, however we consider that the lack of this information could be high light as a limitation. We included a comment about this limitation in the discussion of the current version of the manuscript.

In respect to immunosuppression, we were especially interested in the use of high doses of steroids or lymphocyte depleting antibody to treat acute rejection until 3 months from the COVID-19 diagnosis, considered as recent use. 

Recent use of high dose of steroids was recorded for 4.3% (n=73) patients, whereas lymphocyte depleting antibody was recorded for 2.8% (n=47).

These results were embodied in the results, as following:

“Use of high dose of steroids and lymphocyte depleting antibody to treat acute rejection until 3 months before the COVID-19 diagnosis were recorded for 4.3% (n=73) and 2.8% (n=47), respectively.”

Reviewer #2 comments:

What was the time between symptoms and hospitalization in those hospitalized? This data would be useful for clinicians to know when patients are at the highest risk for hospitalization.

Author comments:

Thank you for your very important suggestion.

The time between symptoms and hospitalization was 7.0 (4.0; 10.0) days. This information was included in the results of the current version of the manuscript, as following:

“About two-thirds of the patients (65.1%) were hospitalized for clinical management, 7.0 (4.0; 10.0) days after the beginning of symptoms.”

Reviewer #2 comments:

Nosocomial infections were high. Did these occur at the time of transplantation or while admitted for another cause?

Author comments:

Thank you for the opportunity to clarify this information: 137 patients had the nosocomial infection as the source of infection. Among them, in 16 (11.7%) the infection occurred during the hospitalization when the transplantation was performed. We embodied this information in the current version of the manuscript.

Reviewer #2 comments:

The authors mention deep social inequality in Brazil. Do they have any information on socioeconomic status in the current study?

Author comments:

The socioeconomic status was not available in the present study. We agree that it is a very important point, and it should be addressed in future investigations. We are including this statement in the discussion.

Reviewer #2 comments:

As the authors have a good number of patients perhaps they analyze the following:

a. Outcomes in patients transplanted early (<1 year) vs late (>1 year)

b. Outcomes in patients without patient reduction in immunosuppression versus no change. It would be interesting to know if early reduction in immunosuppression can help avoid hospital admission or perhaps it makes no difference.

c. Can the authors perform a comparison between hospitalized and non-hospitalized, as well as survivors vs. non-survivors. Were there any other characteristics besides AKI that predicted a higher risk for adverse outcome (for example age, race, co-morbid conditions, regions, or times of disease)?

d. Did any patients have a kidney biopsy? Were there any episodes of rejection?

Author comments:

Point a:

Thank you for your suggestion.

The outcomes were compared according to the time of transplantation, which was dichotomized in ≤1 year and >1 years. These results were included in the supplementary table 2 in the current version of the manuscript.

The hospitalization rate was higher among patients who had the time after transplantation shorter than 1 year (71.3% vs. 64.0%, P=0.031), as well as the requirement for RRT (30.6% vs. 22.1, P=0.005).

The following sentence was included in the results in the current version:

“The outcomes were compared according to the time of transplantation: longer or shorter than 1 year (supplementary table 2). The hospitalization rate was higher among patients who had the time after transplantation shorter than 1 year (71.3% vs. 64.0%, P=0.031), as well as the requirement for RRT (30.6% vs. 22.1, P=0.005).”

Point b:

We totally agree this is a very relevant point. It would be interesting to seek if early management could impact in the outcomes, such as hospitalization. However, the time when the immunosuppression was reduced or withdrawn was not available.

Point c:

Thank you for this suggestion. We compared the variables according to the hospitalization (yes vs. no) and death (survivors vs. non-survivors). The continuous variables were compared by Mann-Whitney U test and the categorical were compared by X2 or Fisher test. We are including these results in the current version of the manuscript. 

In the supplementary table 3 is shown the comparison between patients who were hospitalized versus those who were not. The variables that reached a P-value ≤ 0.10 in the univariable comparison were selected for the multivariable modeling. The multivariable analysis was performed by the binary logistic regression. The final results are shown in the table 4. 

In summary, the variables related with the probability of hospitalization were age, hypertension, previous cardiovascular disease, recent use of high dose of steroid, and fever, dyspnea, diarrhea, and nausea/vomiting as COVID-19 symptoms. On the other hand, the variables that reduced the probability of hospitalization were time of COVID-19 symptoms, and nasal congestion, headache, arthralgia and anosmia as COVID-19 symptoms.

In the supplementary table 4 is shown the comparison between patients who survived versus those who did not. Similarly, the variables that reached a P-value ≤ 0.10 in the univariable comparison were selected for the multivariable modeling. The multivariable analysis was performed by the binary logistic regression. The final results are shown in the table 5.

The variables related with the probability of death within 90 days after COVID-19 were age, time after transplantation, presence of hypertension, previous cardiovascular disease, use of tacrolimus and mycophenolate as the maintenance immunosuppression, recent use of high dose of steroids, and dyspnea as COVID-19 symptom. On the other hand, the variables that reduced the risk of death were time of symptoms, and headache and anosmia as COVID-19 symptoms.

Point d:

Unfortunately, information about kidney biopsy after the COVID-19 diagnosis was not available. We consider that this lack of information is a limitation to understand the results about acute kidney injury, and we included this limitation in the discussion.

---

## [Decision Letter · Decision Letter 1]

5 Jul 2021

High mortality among kidney transplant recipients diagnosed with coronavirus disease 2019: Results from the Brazilian multicenter cohort study.

PONE-D-21-05548R1

Dear Dr. Requião-Moura,

We’re pleased to inform you that your manuscript has been judged scientifically suitable for publication and will be formally accepted for publication once it meets all outstanding technical requirements.

**The revised version of the manuscript is definitely improved. The authors have properly addressed all the critiques and comments raised by the Reviewers.**

Kind regards,

Giuseppe Remuzzi

Academic Editor

PLOS ONE

Additional Editor Comments (optional):

Reviewers' comments:

Reviewer's Responses to Questions

**Comments to the Author**

1. If the authors have adequately addressed your comments raised in a previous round of review and you feel that this manuscript is now acceptable for publication, you may indicate that here to bypass the “Comments to the Author” section, enter your conflict of interest statement in the “Confidential to Editor” section, and submit your "Accept" recommendation.

Reviewer #2: All comments have been addressed

2. Is the manuscript technically sound, and do the data support the conclusions?

Reviewer #2: (No Response)

3. Has the statistical analysis been performed appropriately and rigorously? 

Reviewer #2: (No Response)

4. Have the authors made all data underlying the findings in their manuscript fully available?

Reviewer #2: (No Response)

5. Is the manuscript presented in an intelligible fashion and written in standard English?

Reviewer #2: (No Response)

6. Review Comments to the Author

Reviewer #2: (No Response)

7. PLOS authors have the option to publish the peer review history of their article (what does this mean?). If published, this will include your full peer review and any attached files.

Reviewer #2: No

---

## [Editor Report · Acceptance letter]

12 Jul 2021

PONE-D-21-05548R1 

High mortality among kidney transplant recipients diagnosed with coronavirus disease 2019: Results from the Brazilian multicenter cohort study. 

Dear Dr. Requião-Moura:

I'm pleased to inform you that your manuscript has been deemed suitable for publication in PLOS ONE. Congratulations! Your manuscript is now with our production department. 

Kind regards, 

on behalf of

Prof. Giuseppe Remuzzi 

Academic Editor

PLOS ONE